

# S-Swin Transformer: simplified Swin Transformer model for offline handwritten Chinese character recognition

Yongping Dan, Zongnan Zhu, Weishou Jin and Zhuo Li

School of Electronic Information, Zhongyuan University of Technology, Zhengzhou, Henan, China

## ABSTRACT

The Transformer shows good prospects in computer vision. However, the Swin Transformer model has the disadvantage of a large number of parameters and high computational effort. To effectively solve these problems of the model, a simplified Swin Transformer (S-Swin Transformer) model was proposed in this article for handwritten Chinese character recognition. The model simplifies the initial four hierarchical stages into three hierarchical stages. In addition, the new model increases the size of the window in the window attention; the number of patches in the window is larger; and the perceptual field of the window is increased. As the network model deepens, the size of patches becomes larger, and the perceived range of each patch increases. Meanwhile, the purpose of shifting the window's attention is to enhance the information interaction between the window and the window. Experimental results show that the verification accuracy improves slightly as the window becomes larger. The best validation accuracy of the simplified Swin Transformer model on the dataset reached 95.70%. The number of parameters is only 8.69 million, and FLOPs are 2.90G, which greatly reduces the number of parameters and computation of the model and proves the correctness and validity of the proposed model.

## INTRODUCTION

Chinese characters have a long history as the most widely used script in China and even globally. Currently, handwritten Chinese character recognition has been studied by researchers for more than 50 years (*Dai, Liu & Xiao, 2007*). Chinese characters have the characteristics of a large number of categories (refer to the GB2312-80 standard; there are 6763 categories of commonly used Chinese characters), many similar glyphs, and a diversity of writing styles, which makes the recognition of handwritten Chinese characters one of the most challenging tasks in the field of pattern recognition.

Chinese character recognition includes both types of printed Chinese character recognition and handwritten Chinese character recognition (HCCR). Handwritten Chinese character recognition can be subdivided into offline HCCR and online HCCR according to different data collection methods. Online handwritten Chinese character recognition mainly refers to the recognition of Chinese characters written on the electronic screen.

Corresponding author
Yongping Dan, 6100@zut.edu.cn, 420076822@qq.com

During the recognition process, the timing information of Chinese writing and the relative position information of strokes are recorded. The offline handwritten Chinese character recognition object is the offline handwritten Chinese character image collected by the scanner or camera. These Chinese character images are affected by distortion, imaging equipment limitations, and other reasons, and there are few effective features that can be used. At the same time, the recognition of offline handwritten Chinese characters is more difficult due to the complex structure of Chinese characters and the irregularities in the writing process. The recognition of offline handwritten Chinese characters has great limitations. Therefore, research in this field still has very important significance and application value.

In online HCCR, the trajectory of the pen tip movement while writing on a dedicated digitizer or personal digital assistant is recorded and analyzed in real-time. This trajectory information is automatically converted into linguistic information to be expressed by the recognition (*Liu, Jaeger & Nakagawa, 2004*). Today, online HCCR technology is very mature and has been widely used in pen input devices, writing pads, computer-aided education, smartphones, and other fields. In offline HCCR, the text in character images is automatically converted into alphabetic codes and classified into different categories. Offline HCCR has essential applications in accessible reading for people with disabilities, automatic document entry, mail sorting (*Liu, Koga & Fujisawa, 2004*), signature checking, banknote processing, and document recognition. Due to its high practicality, the study of offline HCCR has received a lot of attention, and many methods have been proposed to improve the recognition accuracy. However, offline HCCR has specific difficulties. The recognition accuracy still needs to be improved, and the technology still needs to be more perfect.

The traditional offline HCCR system mainly includes three steps: data preprocessing, feature extraction, and classification recognition (*Jin, Zhong & Yang, 2016*). The conventional recognition methods are mature enough and have reached their accuracy limit, and an innovative approach is needed to break this limit. The emergence of deep learning provides new ways to break this limit and provides researchers with new ideas. Therefore, many meaningful theories and algorithms have been proposed by scholars to solve this problem. Some existing network models, such as Convolutional Neural Networks (CNNs) (*Cun et al., 1990*), Deep Belief Networks (DBN) (*Hinton & Salakhutdinov, 2006*), and Deep Recurrent Neural Networks (DRNN) (*Gers & Schmidhuber, 2001*), consider the basic structural features of Chinese characters. And these approaches are practical in offline HCCR tasks.

The Transformer (*Vaswani et al., 2017*), a novel neural network, was first applied to natural language processing (NLP) tasks, such as machine translation and English constituency analysis tasks, and achieved significant improvements in results. In the field of computer vision, Transformer-based models mainly use the key module self-attention mechanism to extract intrinsic features and show great potential in artificial intelligence applications, such as high-resolution image synthesis (*Dalmaz, Yurt & Ukur, 2021*), object detection (*Carion et al., 2020*), classification (*Yuan et al., 2021*), segmentation (*Zheng et al., 2021*), image processing (*Lin et al., 2021*), and re-identification (*Luo et al., 2020*).

Furthermore, in vision applications, CNNs have previously been considered the fundamental component, but now the transformer shows that it will be a potential replacement for CNNs. At present, lightweight convolutional neural networks are more common (*Dong et al., 2022*). There are a few lightweight models based on Transformer, and most of them are complex models with a large number of parameters. In the near future, everyone expects Transformer to have the same status as CNN in the computer field. Lightweight neural models are bound to become a research trend and drive their deployment on mobile devices, such as Raspberry Pi, FPGA, and human–computer interaction robots (*Yang, Chew & Liu, 2021*).

The rest of this article is structured as follows. The Related Works section briefly reviews the work related to the Transformer model and HCCR. The Methods section introduces the internal structure and working principles of the method in detail. The Experimental and Results sections clearly describe the experimental procedure and experimental results. The Conclusions section concludes the article and represents the direction of future work.

The main contributions of our work are as follows:

• This article proposes a simplified Swin Transformer (S-Swin Transformer) model for handwritten Chinese character recognition. This model simplifies the relatively complex model structure by removing some encoder layers. Experiments show that this method can effectively reduce the number of parameters and calculation of the model, and the recognition accuracy is also satisfactory.

• The S-Swin Transformer model increases the size of the window in the window attention, changing the window size from the original $7 \times 7$ to $14 \times 14$. With the deepening of the network model, the patch keeps getting bigger. The perceptual range of each patch increases and contains more information. In addition, the experimental results show that the validation accuracy is slightly improved when the window is increased to $14 \times 14$.

## RELATED WORKS

### Offline HCCR

Offline HCCR has been considered a complex and challenging task for academics owing to its wide range of character categories, diverse writing styles, and complex text structure. However, with the rapid development of technology and the economy, deep learning models (*Lecun, Bengio & Hinton, 2015*) have slowly entered the public perspective. Through the continuous efforts of researchers in research and innovation, deep learning models are widely used in the field of computer vision with great success and far-reaching impact.

At present, handwritten Chinese character recognition methods are starting to make the transition from using convolutional neural networks (CNNs) instead of traditional Chinese character recognition methods (*Lecun & Bottou, 1998*). Deep learning-based approaches are becoming increasingly popular in handwritten Chinese character recognition. In the literature (*Liu et al., 2013*), the authors mention that the best-performing methods in handwritten Chinese character recognition are currently based on deep neural network models. Among them, the multi-column deep learning network (MCDNN) is considered to be the first convolutional neural network (CNN) model to be successfully applied in

HCCR (*Cirean & Schmidhuber, 2013*). The multi-column deep learning network consists of multiple CNNs, and the final recognition accuracy is comparable to human performance. In the offline HCCR competition held by ICDAR in 2013, Fujitsu's team achieved 94.77% recognition accuracy using a CNN-based model and won first place in the competition (*Yin et al., 2013*). A CNN-based framework for handwritten character recognition was proposed by *Li et al. (2015)*, who used an appropriate sample generation, training scheme, and CNN network structure with a recognition error rate of only 3.21% on the CASIA dataset. In 2016, Zhang et al. combined the traditional normalized collaborative directional decomposition feature map (direct map) with a deep convolutional neural network (convert), which achieved higher recognition accuracy for both online and offline HCCR (*Zhang, Bengio & Liu, 2016*). A network that is well balanced in terms of speed, scale, and performance was proposed by *Li et al. (2018)*. Their cascaded single CNN model classifies character images on the CPU in 6.93 ms, with a recognition accuracy of 97.11% and only 3 M of storage space required. An offline handwritten Chinese character recognition method based on a deep convolutional generative adversarial network (DCGAN) and improved GoogleNet was proposed by *Li, Song & Zhang (2018)*, which is capable of repairing and recognizing obscured characters. The proposed method was evaluated on the extended CASIA-HWDB1.1. The experimental results show that the method can obtain a higher repair rate and recognition accuracy than most methods. A writing style adversarial network (WSAN) structure was proposed by *Liu et al. (2019)*. This network contains three parts: a feature extractor, a character classifier, and a writer classifier. The authors use a feature extractor to learn a deep representation of the original image, and then jointly optimize the network by minimizing the loss of the character classifier and maximizing the loss of the writer classifier. The experimental results on CASIA-HWDB1.1 prove that the writing style adversarial network (WSAN) promotes the HCCR results.

## Transformer

Inspired by the significant success of Transformer architectures in the NLP domain (*Devlin et al., 2018*), researchers have recently applied transformers to computer vision(CV) tasks. With the development in recent years, various Transformer variants have been proposed by researchers, which are also known as X-Transformer models. These methods have made good progress in their applications to multiple tasks.

The first convolution-independent Vision Transformer (ViT) model was proposed by *Dosovitskiy et al. (2020)*. This method directly uses the sequence of embedded image blocks as the input to a standard converter, and experiments demonstrate that this model can perform the image classification task excellently. The teacher-student strategy for Transformer was introduced by *Touvron et al. (2020)*. It relies on a distillation token to ensure that the student learns from the teacher through attention. This approach achieved 85.2 % accuracy on the ImageNet-1K dataset. To generate stronger image features, the two-branch transformer was proposed by *Chen, Fan & Panda (2021)*. The model processes large and small patch tokens of two independent branches with different computational complexity, and then fuses these tokens purely through multi-attention. Ultimately, this method has a substantial 2% accuracy advantage over the recent DeiT (*Touvron et al.,*

*2020*) on the ImageNet1K dataset. A new structure for the convolutional vision transformer (CVT) was proposed by *Wu et al. (2021)*. The authors demonstrated that this structure combines the advantages of converters with those of CNNs in image recognition tasks, and the authors validated CVT by conducting extensive experiments. The results showed that the method achieved better performance with fewer parameters and fewer FLOPs on the ImageNet-1k dataset. A simple and effective re-attention(re-attention) method was introduced by *Zhou et al. (2021)*. The authors made minor modifications to the existing ViT model and improved the Top-1 classification accuracy by 1.6% on the ImageNet dataset when training a deep ViT model using 32 transformer blocks.

## METHODS

A simplified Swin Transformer (S-Swin Transformer) model for handwritten Chinese character recognition is proposed in this article. The model simplifies and compresses the Swin Transformer generic framework (*Liu et al., 2021*). The complete S-Swin Transformer model architecture is shown in Fig. 1 (except for Stage 4). First, compared with the Swin Transformer structure, the S-Swin Transformer model proposed in this article has only three "Stages" and one fewer "Stage 4". This model can effectively reduce the number of parameters of the model. Then, we also set the attention window of the proposed model to $14 \times 14$. The purpose is to enable more information exchange between patches in the window. Finally, it is proved experimentally that the proposed new model can not only effectively reduce the number of parameters of the model but also reduce the FLOPs, and the experimental accuracy achieves the expected results.

### S-Swin Transformer model

As shown in Fig. 1. First, input any 3-channel (RGB) image $x \in H \times W \times C$, where H, W, and C represent the height, width, and number of channels of the image, respectively. The images were sliced into non-overlapping image blocks by the patch partition module. Each non-overlapping image block was regarded as a token. Then these tokens are fed into "Stage 1". "Stage 1" contains a linear embedding layer and an S-Swin Transformer block layer. The standard size $224 \times 224$ image is sliced into $H/4 \times W/4$ small image blocks of $4 \times 4$ pixels. Each image block has a feature dimension of 48. The image blocks are mapped to C dimensions after a linear embedding layer. Also, unlike in the standard Transformer, the model uses an attention module based on the shift window, and no other structural layers have changed. The S-Swin Transformer block structure is shown in Fig. 2. The S-Swin Transformer block structure of each stage is composed of two successive connected Transformer encoders. The difference is that the first one contains a core module windowed multi-head self-attention (W-MSA) and a multilayer perceptron (MLP). The second one is composed of the core module shifting window multi-head self-attention (SW-MSA) and MLP. Finally, residual connections are used after each module. Moreover, LayerNorm (LN) layers are applied before each W-MSA, SW-MSA, and MLP module.

"Stage 2" consists of a patch merging layer and an S-Swin Transformer Block layer. With the deepening of the network model, the patch merging layer merges four adjacent $4 \times 4$ image blocks, then the patch pixels are adjusted to $8 \times 8$ in "Stage2", and each $8 \times 8$

**Peer**J Computer Science

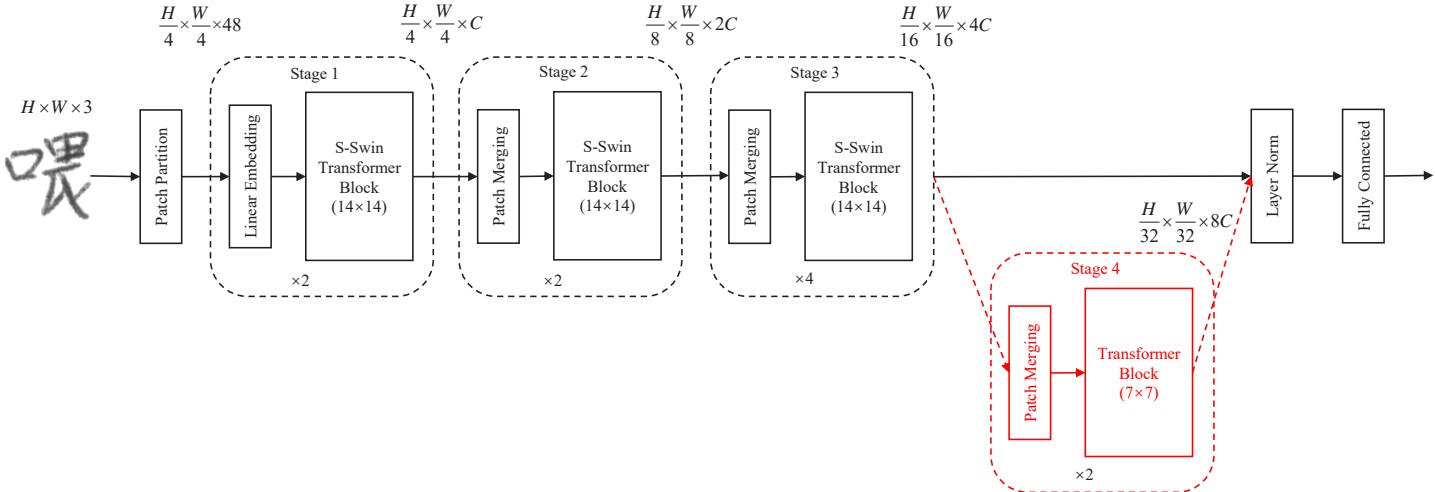

**Figure 1** Complete S-Swin Transformer model architecture (except for the Stage 4 part).

patch is considered as a token. The image has a total of $H/8 \times W/8$ tokens, and the output dimension is mapped to 2C. The patch merging layer and the S-Swin Transformer Block layer form "Stage 3". Like the composition structure of "Stage 2", the patch merging layer merges four adjacent $8 \times 8$ image blocks, and then the patch pixels are resized to $16 \times 16$ in "Stage 3". Each $H/16 \times W/16$ patch is regarded as a token; the picture has a total of $H/16 \times W/16$ tokens, and the output dimension is mapped to 4C. In addition, the Swin Transformer Block layer has been doubled compared to "Stage 2". Finally, the patch size change in the model is shown in Fig. 3.

## Window of attention

The multi-head attention module in the standard Transformer architecture processes images with global self-attention, where the correlation between a token and all other tokens is calculated, which directly causes the model to be computationally intensive. The S-Swin Transformer model proposes window attention and shifting window attention, which are used consecutively together, as shown in Fig. 2. The role of the attention window is to calculate the self-attention for the patches contained in the window species in the set window size. If the window is set to $N \times N$, it means that there are $N \times N$ patches in the window. N is set at 14 in the S-Swin Transformer model. Compared to computing the attention relationship among all patches, window attention greatly reduces the computational effort. However, the window attention module lacks the information interaction between windows, which will lead to the poor modeling ability of the model. The purpose of shifting window attention is to address the lack of interrelationships between a window and patches within other windows. It allows more patches to be connected through self-attention calculations, enhancing the modeling capability of the model. The continuous S-Swin Transformer block layer is calculated as follows.

$$\hat{z}_i = W - MSA(LN(z_{i-1})) + z_{i-1} \tag{1}$$

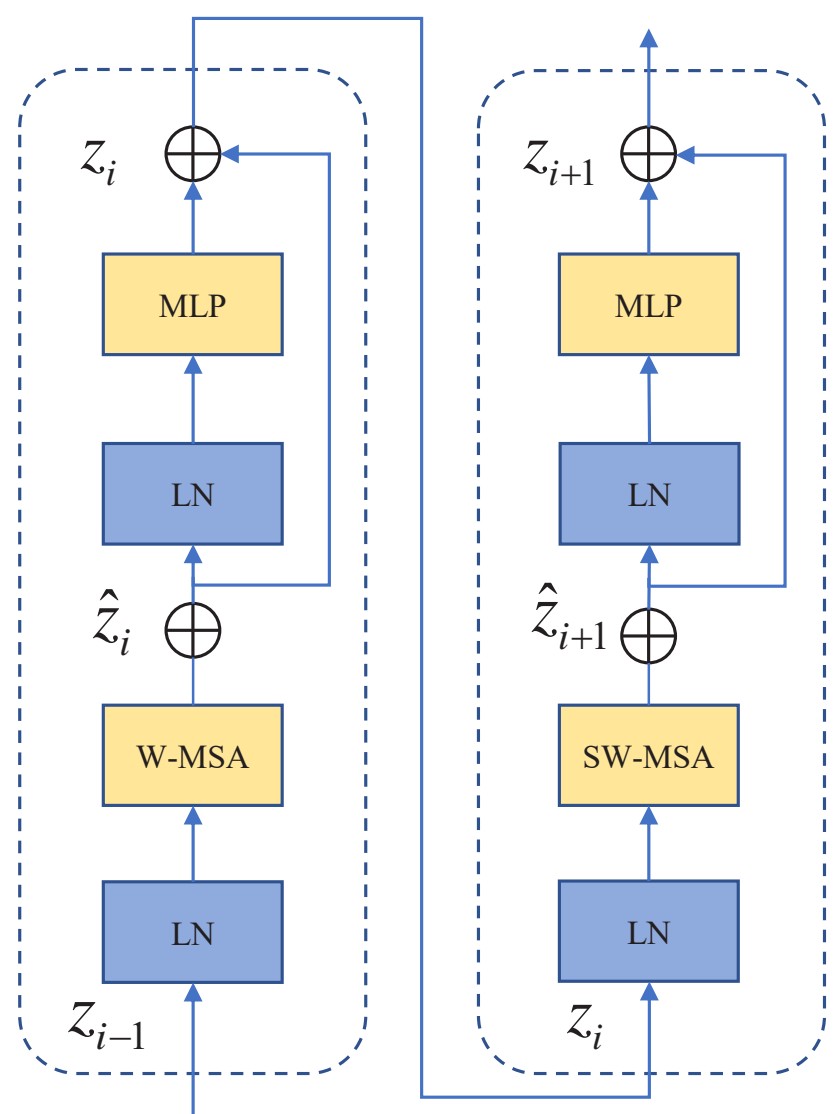

**Figure 2** S-Swin Transformer block layer.

$$z_i = MLP(LN(\hat{z}_i)) + \hat{z}_i \qquad (2)$$

$$\hat{z}_{i+1} = SW - MSA(LN(z_i)) + z_i \qquad (3)$$

$$z_{i+1} = MLP(LN(\hat{z}_{i+1})) + \hat{z}_{i+1} \qquad (4)$$

## Multi-head self-attention mechanism

The multi-head attention structure is the most central component of the S-Swin Transformer model, which consists of four layers: a linear layer, a self-attention layer,

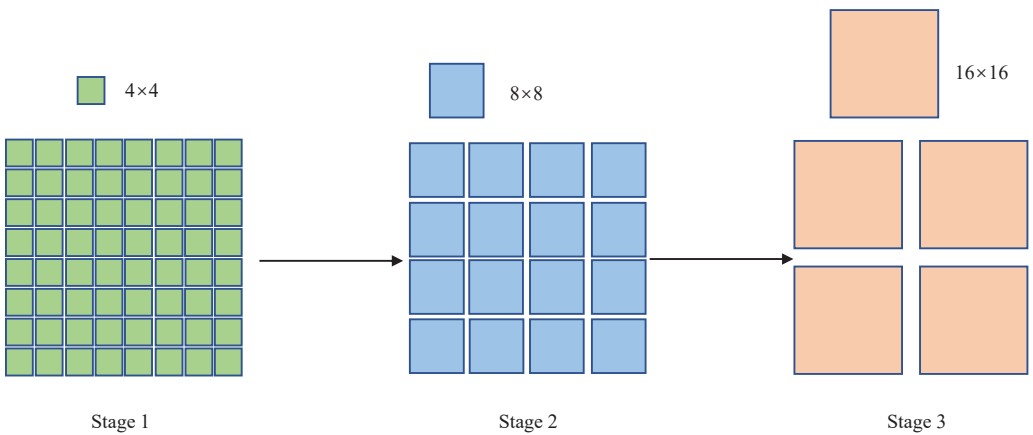

**Figure 3   The patch size change in the model.**

a multiple-attention-head splicing layer, and a linear layer. The detailed structure is shown in Fig. 4. In window attention, attention weights are obtained by calculating the dot product of query (Q), key (K), and value (V). Firstly, the input sequence vector is multiplied with three learnable vector matrices ($W_q$, $W_k$, $W_v$) to obtain Q, K, and V. Q is multiplied by the transpose dot product of all K and then divided by the square root of the K dimensions and fed to the Softmax function. Finally, the output value of the Softmax function is multiplied by V to obtain the attention weights. The calculation procedure is shown in Eq. (5). In addition, the softmax function is also known as the normalized exponential function. The output value of the function ranges from [0, 1] and adds up to 1. The specific procedure is shown in Eq. (6).

$$\text{Attention}(Q, K, V) = \text{Softmax}(\frac{QK^T}{\sqrt{d}})V \tag{5}$$

$$\text{Softmax}(z_l) = \frac{e^{z_l}}{\sum_{c=1}^{C} e^{z_c}} \tag{6}$$

The number of heads h is different for each stage in the model. In stages 1, 2, and 3, h is set to 3, 6, and 12, respectively. The attention value of each head is calculated, and finally, they are spliced together. The detailed representation is shown in Eqs. (7) and (8).

$$\text{head}_j = \text{Attention}(QW_j^q, KW_j^k, VW_j^v) \tag{7}$$

$$\text{MultiHead}(Q, K, V) = \text{Concat}(\text{head}_1, \cdots, \text{head}_h)W^O \tag{8}$$

## EXPERIMENTAL AND RESULTS

### Dataset

The famous offline handwritten Chinese character public dataset CASIA-HWDB1.1 (*Liu et al., 2011*) contains 3755 categories. Each of these characters is written by 300 authors,

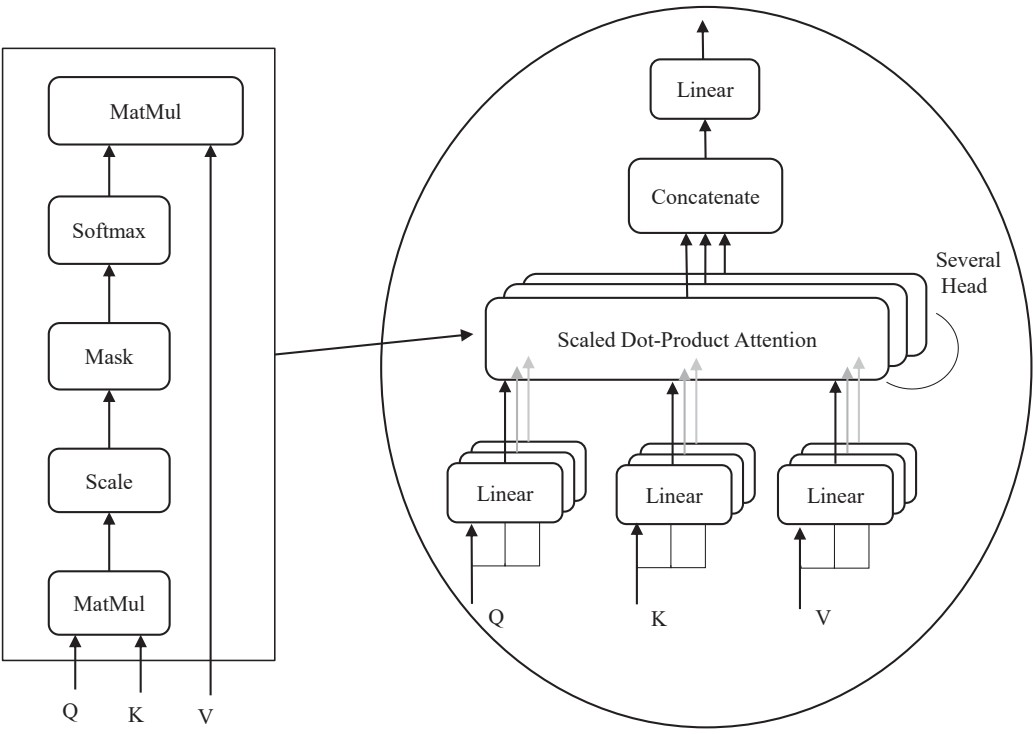

**Figure 4** (A) Multi-head attention. (B) Self-attention process.

**Table 1** Dataset characteristics.

| Dataset | Classification | Total images | Training ratio |
| --- | --- | --- | --- |
| T-HWDB1.1 | 300 | 104105 | 80% |

and each image is represented as an 8-bit grayscale image. The total number of images in the HWDB1.1 dataset is huge, nearly 900,000. However, according to the previous experience of researchers, the size of the data has both advantages and disadvantages. Using more training data can lead to higher recognition accuracy. On the contrary, too much training data can affect the training efficiency of the model. The more training data is used, the higher recognition accuracy can be obtained. Conversely, the training efficiency of the model is also reduced due to a large amount of training data. Most importantly, due to the limitations of the laboratory equipment, 300 categories were randomly selected from CASIA-HWDB1.1 to compose the final dataset used for the experiment. This dataset is named T-HWDB1.1. The dataset has a total of 104,105 images. T-HWDB1.1 is also randomly divided into an 80% training set and a 20% validation set.The specific dataset characteristics are shown in Table 1. Additionally, Fig. 5 represents sample examples of individual categories in the dataset. And Fig. 6 represents the diversity of writing, where each column represents that the same character is written by different people.

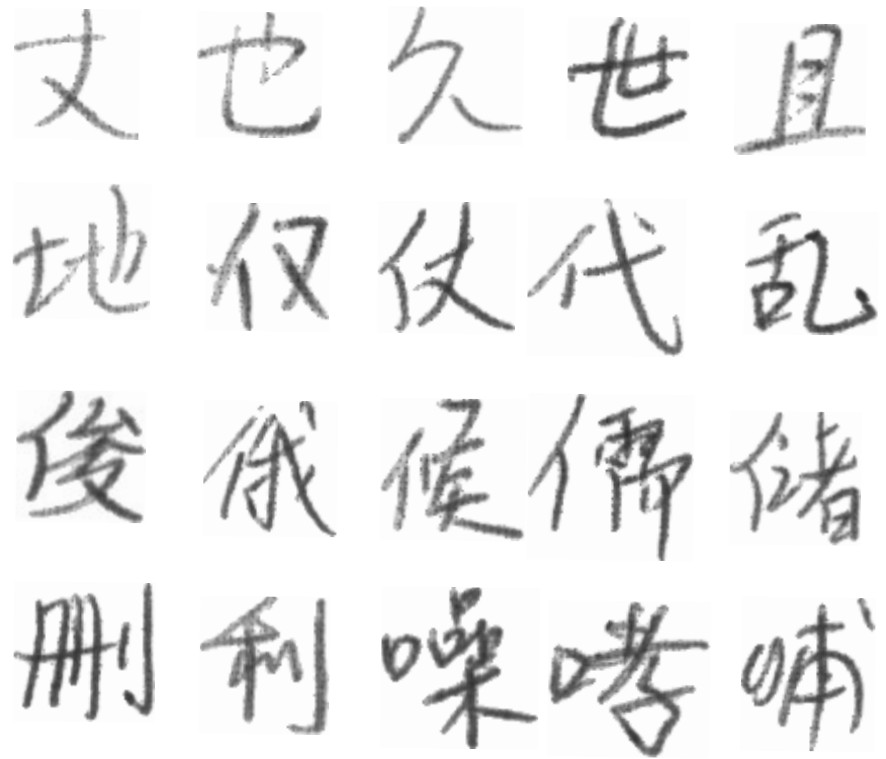

**Figure 5** Sample examples of individual categories in the dataset.

**Figure 6** Each column represents the same character written by different people.

**Table 2  Parameter settings.**

| Description | Value |
| --- | --- |
| Input | 224×224 |
| Learning rate | 0.0001 |
| Batch size | 8 |
| Dropout | 0.1 |
| Epochs | 150 |

**Table 3  AlexNet and VGG16 parameter settings.**

| Models | Parameters | | | |
| --- | --- | --- | --- | --- |
| | Learning rate | Batch size | Dropout | Epochs |
| AlexNet (*Krizhevsky, Sutskever & Hinton, 2012*) | 0.001 | 8 | 0.1 | 200 |
| VGG16 (*Simonyan & Zisserman, 2014*) | 0.001 | 8 | 0.1 | 200 |

## Experimental settings

First, for a fair and efficient comparison experiment, the hyperparameters of all models in the experiment are set to constant values. The detailed experimental setup parameters are shown in Table 2. During the training and validation process, all input images are resized to $224 \times 224$. The batch size is set to 8, and the number of training iterations is 150. The learning rate is set to 0.0001, and the window size is set to 7 or 14. Dropout regularization is also used during training, and the dropout parameter is set to 0.1. The purpose of Dropout is to effectively avoid overfitting problems during training and to increase the generalization ability of the model. Then, this article uses PyTorch to implement the network algorithm flow. All of the experiments are conducted on a computer with a 3.00 GHz Intel (R) Core (TM) i7-9700 processor, $2 \times 8$ GB of RAM, and a GeForce RTX 2060 graphics card with 6GB of video memory.

The model S-Swin Transformer proposed in the article and the AlexNet (*Krizhevsky, Sutskever & Hinton, 2012*) and VGG16 (*Simonyan & Zisserman, 2014*) networks were compared experimentally on the same dataset T-HWDB1.1. The detailed parameter settings of the AlexNet and VGG16 networks are shown in Table 3. AlexNet is an 8-layer deep network with five convolutional layers and three fully connected layers. The advantage of the convolutional layer is that it extracts effective features with a small number of parameters. Alexnet uses max pooling to avoid the blurring effect of average pooling. The VGG16 network consists of 13 convolutional layers and three fully connected layers. All convolutional layers use 3 * 3 convolution kernels. The role of convolutional layers and pooling layers is to extract image features. The final fully connected layer is mainly responsible for completing the recognition and classification.

## Experimental results

Detailed experimental results are shown in Table 4, presenting the model shifting attention window size, verification accuracy, number of parameters, and FLOPs, respectively. From the experimental results, both AlexNet and VGG16 achieve the highest recognition accuracy of 95.10% on T-HWDB1.1. When compared to the best verification accuracy of the S-Swin

**Table 4  Experimental results (Validation Accuracy, Number of Parameters, FLOPs).**

| Model | Window_size | Accuracy (%) | Parameter (M) | FLOPs (G) |
|---|---|---|---|---|
| AlexNet (*Krizhevsky, Sutskever & Hinton, 2012*) | | 95.10 | 15.19 | 0.30 |
| VGG16 (*Simonyan & Zisserman, 2014*) | | 95.10 | 135.48 | 15.40 |
| Swin transformer | 7×7 | 95.10 | 27.70 | 4.30 |
| S-Swin transformer | 7×7 | 95.40 | 8.69 | 2.90 |
| Swin transformer | 14×14 | 95.40 | 27.70 | 4.30 |
| S-Swin transformer | 14×14 | 95.70 | 8.69 | 2.90 |

Transformer (window of $14 \times 14$), the accuracy is 0.60% lower. The parameters of AlexNet and VGG16 are 6.50 million and 126.79 million more than those of the model proposed in this article, respectively. In addition, their FLOPs are 0.30G and 15.40G, respectively. Furthermore, when the Swin Transformer model is used and the shift attention window size is set to $7 \times 7$, the final result of the experiment achieves a validation accuracy of 95.10%. At this time, the parameters are 27.70 million. The FLOPs are 4.30G. Using the simplified S-Swin Transformer model proposed in this article, experiments are conducted on the dataset with 150 iterations when setting the shift attention window size to $7 \times 7$. The final experimental validation accuracy reaches 95.40%, the number of parameters is significantly reduced to only 8.69 million, and the FLOPs are only 2.90G. Compared with the Swin Transformer model with an attention window size of $7 \times 7$, the verification accuracy increases by 0.30%, the parameter sizes decrease by 19 million, and the FLOPs decrease by 1.40G. In addition, the training results achieve 95.40% validation accuracy when the attention window size of the Swin Transformer model is set to $14 \times 14$. When compared to a window size of $7 \times 7$ validation accuracy increases by 0.30% while parameter size and FLOPs remain constant.

With 8.69 million and 2.90G of parametric quantities and FLOPs, respectively, the final validation accuracy reaches 95.70% when the S-Swin Transformer model window size is changed to $14 times 14$. Compared with the S-Swin Transformer model with the attention window set to $7 \times 7$, the number of parameters and the size of FLOPs were the same, while the verification accuracy increases by 0.30%. Meanwhile, compared with the Swin Transformer model with the attention window set to $14 \times 14$, the number of parameters is reduced by 19 million, and the FLOPs are reduced by 1.40G.

Finally, the results of the entire verification process are recorded and implemented by the visualization tool tensorboard. The curves of validation accuracy and iteration number of the proposed S-Swin Transformer model with a $14 \times 14$ window on the dataset T-HWDB1.1 are shown in Fig. 7. The vertical and horizontal axes represent the validation accuracy and the number of experimental iterations, respectively. When the experiment is iterated 300 times, the verification accuracy reaches 95.70%. In addition, the change curve of the loss function with the number of iterations during the verification process is shown in Fig. 8. When the number of iterations is set to 300, the loss function value is minimized. The most important thing is that after a certain number of experimental iterations, the

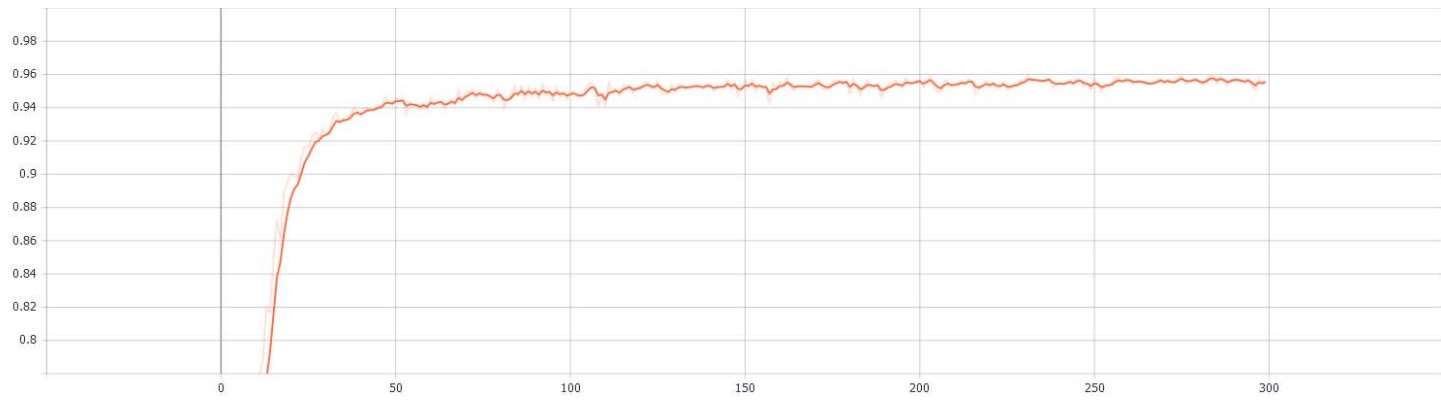

**Figure 7** The validation accuracy and iteration number curves.

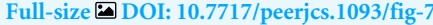

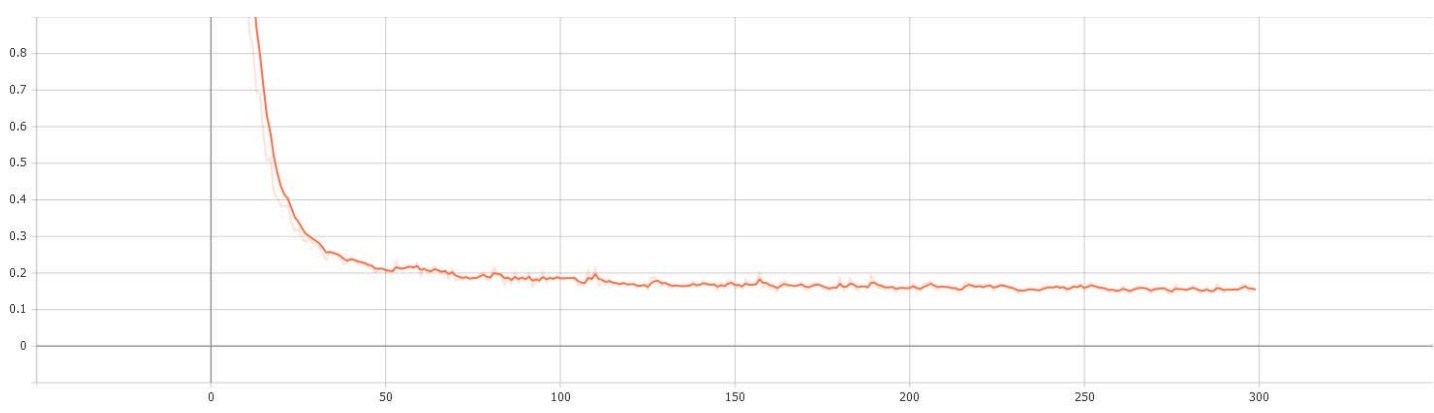

**Figure 8** The change curve of the loss function with the number of iterations in the verification process.

verification accuracy will not increase with the increase in the number of iterations but will remain near optimal.

## CONCLUSIONS

A simplified Swin Transformer (S-Swin Transformer) model for handwritten Chinese character recognition is proposed in this article. In addition, this article also explores the effect of window size on validation accuracy by varying the size of the moving attention window of the proposed S-Swin Transformer model. Several sets of comparison experiments were conducted on the dataset T-HWDB1.1. According to the experimental results, the simplified Swin Transformer model (with a window size of $14 \times 14$) performed best on the dataset, with an accuracy of 95.70%. The method not only ensures the recognition classification accuracy but also dramatically reduces the number of parameters and FLOPs, with only 8.69 million parameters and 2.90G FLOPs, solving the problem that the model requires a large amount of computation. Moreover, the experimental results after changing the window size of the S-Swin Transformer model showed that the window

size led to a weak change in the validation accuracy. When the shifting attention window is set to $14 \times 14$, the validation accuracy is 0.30% higher than when the attention window is set to $7 \times 7$. In conclusion, the experimental results verified the correctness and validity of the proposed method.

In future research, knowledge distillation will be introduced in the model to make the model parameters and computation less complex, allowing the model to be more easily ported to embedded devices. With the continuous development of society, transformer-based models will have broader application prospects, such as defect recognition (*Zhao et al., 2021*; *Hu et al., 2021*) and weakly supervised target detection (*Sun et al., 2018*).

## ACKNOWLEDGEMENTS

Thanks to CASIA-HWDB dataset provided by the Institute of Automation of Chinese Academy of Sciences. The offline Chinese handwriting databases, built by the CASIA, are released for academic research free of cost under an agreement.

### Funding
The authors received no funding for this work.

### Competing Interests
The authors declare there are no competing interests.

### Author Contributions
- Yongping Dan conceived and designed the experiments, performed the experiments, analyzed the data, authored or reviewed drafts of the article, writing—review and editing, and approved the final draft.
- Zongnan Zhu performed the experiments, analyzed the data, performed the computation work, prepared figures and/or tables, writing—original draft preparation, and approved the final draft.
- Weishou Jin performed the experiments, performed the computation work, prepared figures and/or tables, validation, and approved the final draft.
- Zhuo Li analyzed the data, performed the computation work, prepared figures and/or tables, visualization, and approved the final draft.

### Data Availability
The code is available in the Supplemental File.

The data is available at figshare: Zhu, Zongnan (2022): T-HWDB1.1.zip. figshare. Dataset. https://doi.org/10.6084/m9.figshare.20546934.v1.

The data came from a third party database available at CASIA Online and Offline Chinese Handwriting Databases

(CASIA-HWDB): http://www.nlpr.ia.ac.cn/databases/handwriting/Home.html.

The offline Chinese handwriting databases, built by the CASIA, are available for academic research free of cost under an agreement at: http://www.nlpr.ia.ac.cn/databases/handwriting/Offline_database.html.

## Supplemental Information

Supplemental information for this article can be found online at http://dx.doi.org/10.7717/peerj-cs.1093#supplemental-information.

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
