# Peer review of "S-Swin Transformer: simplified Swin Transformer model for offline handwritten Chinese character recognition"

_PeerJ Computer Science, doi:10.7717/peerj-cs.1093_

## Round 0.1 · original submission · Minor Revisions

The topic is interesting and the contribution is acceptable. However, the novelty of the paper should be further highlighted as the current presentation cannot distinguish the results from the existing ones. The investigated problem should be described formally with more details and the typos need to be corrected before resubmitting the revised version.

Reviewer 1 ·

Basic reporting

This manuscript studies a simplified Swin Transformer model for oÿine handwritten Chinese character recognition. The authors considered three hierarchical stages and managed to enhance the information interaction between the window and the window. It is a topic of interest in pattern recognition and computer vision. However, the contributions are not clearly indicated, and there are some minor issues that need to be considered and addressed.

1. The original contributions of this paper should be emphasized compared with existing works.
2. There are some typos and grammatical errors in the manuscript. It is strongly suggested that the whole work be carefully checked by someone who has expertise in technical English writing.
3. Some sentences are too long to follow, it is suggested to break them down into short but meaningful ones to make the manuscript readable.
4. Some of the figures are difficult to observe and read. Please use vector-based images to improve the readability.

Experimental design

1. There are many user-defined parameters in the proposed model. How to select and optimize these parameters?
2. The authors compare the performance of their approach with some of the latest approaches, e.g., AlexNet, VGG16, etc. However, these approaches are not properly cited in the paper.
3. In Figure 7 and Figure 8, it is not clear what are the performances of validation accuracy, change of loss function and so on. These should be clearly described and analysed in the experiment results analysis.
4. Parameter selection and rationale of each method implemented should be indicated in the experiment design.

Validity of the findings

Overall, the authors have presented some interesting results and step forward based on previous works. Conclusions are well stated, and linked to the original research question to support the results. The paper is well organized and logically explained. The quality of the paper can be further improved by addressing the questions and problems raised in this review report.

Reviewer 2 ·

Basic reporting

The article is clearly structured and easy to follow. Comments are listed as follows:
1. What is the different between offline HCCR and online HCCR? Why this work focus on offline HCCR? This should be briefly discussed.
2. There are some obvious formatting/English mistakes in the paper. For example:
Abstract Line11: There is an unnecessary space after the sentence "Transformer shows a good prospect in computer vision"
Introduction Line 63-66: "Section briefly reviews the work related to the Transformer model and HCCR."
Which section?
Such mistakes should be carefully identified.
3. It would be easier to understand if Figure 1 marked both the original Transformer and the Proposed S-Swin Transformer and pointed out the improvements.

Experimental design

In the experimental part, the author introduces the Proposed S-Swin Transformer in detail and conducts comparative experiments. Hyperparameters and database configuration are also specified in detail. However, I still feel the experiments can be made richer. In terms of the recognition success rate, the author only uses the accuracy evaluation to compare the model performance. It will be easier for the reader to understand the performance of the model if it is evaluated using the confusion matrix-based method.

Validity of the findings

Is this study validated using k-fold validation? If not, the reason needs to be briefly stated.

Additional comments

Overall, I think this is a very interesting study. In future work, it will be more practical to realize not only classification but also detection/segmentation.

---

## Round 0.2 · accepted · Accept

All the concerns have been addressed well in the revised version and all the reviewers satisfy the quality of the revision. Therefore, I recommend accepting this manuscript as it is.

Reviewer 1 ·

Basic reporting

The authors have considered the comments and suggestions to improve the manuscript. The contributions have been indicated in the revised manuscript. The typos and grammatical errors have been fixed. Some long sentences have been shortened to improve the readability. Vector-based images have been used to improve readability.

Experimental design

The reviewer's comments have been considered and the revised manuscript is satisfactory. The selection of key user-defined parameters has been clearly stated. The missing references are properly cited in the revised manuscript. Further explanation for Figs 7 and 8 have been added to improve the readability.

Validity of the findings

Rationale and benefit to literature are clearly stated. The conclusions are well stated and linked to the research questions. All underlying data are robust and statistically sound.

Reviewer 2 ·

Basic reporting

The author has made improvements and answered my question.

Experimental design

The author has made improvements and answered my question.

Validity of the findings

The author has made improvements and answered my question.